# Fission Yeast Rho1p-GEFs: From Polarity and Cell Wall Synthesis to Genome Stability

**DOI:** 10.3390/ijms232213888

**Published:** 2022-11-11

**Authors:** Patricia García, Rubén Celador, Jorge Pérez-Parrilla, Yolanda Sánchez

**Affiliations:** Instituto de Biología Funcional y Genómica, CSIC/Universidad de Salamanca and Departamento de Microbiología y Genética, Universidad de Salamanca, C/Zacarías González, s/n, 37007 Salamanca, Spain

**Keywords:** Rho1-GEF, fission yeast, cytokinesis, polar growth, SIN, genomic instability

## Abstract

Rho1p is a membrane-associated protein that belongs to the Rho family of small GTPases. These proteins coordinate processes such as actin remodelling and polarised secretion to maintain the shape and homeostasis of yeast cells. In response to extracellular stimuli, Rho1p undergoes conformational switching between a guanosine triphosphate (GTP)-bound active state and a guanosine diphosphate (GDP)-bound inactive state. Cycling is improved with guanine nucleotide exchange factor (GEF) activity necessary to activate signalling and GTPase activating protein (GAP) activity required for subsequent signal depletion. This review focuses on fission yeast Rho1p GEFs, Rgf1p, Rgf2p, and Rgf3p that belong to the family of DH-PH domain-containing Dbl-related GEFs. They are multi-domain proteins that detect biological signals that induce or inhibit their catalytic activity over Rho1p. Each of them activates Rho1p in different places and times. Rgf1p acts preferentially during polarised growth. Rgf2p is required for sporulation, and Rgf3p plays an essential function in septum synthesis. In addition, we outline the noncanonical roles of Rho1p-GEFs in genomic instability.

## 1. Fission Yeast Rho GTPases

Fission yeast Rho GTPases are highly conserved proteins (~21 Kd) that belong to the family of Rho GTPases and in a higher order to the Ras superfamily of small GTPases. Rho GTPases regulate cytoskeletal dynamics [1], cell polarity establishment, and cell proliferation and invasion [2,3]. Because fission yeasts are walled organisms, most of their known functions in actin and microtubule instruction are intricately linked to traffic control, plasma membrane organisation, and cell wall remodelling (for reviews, see [4,5,6]). More recently, noncanonical functions of fission yeast Rho proteins in signal transduction from the nucleus and other organelles have emerged [7].

*Schizosaccharomyces pombe* (*S. pombe*) has six Rho GTPases: Rho1p, Rho2p, Rho3p, Rho4p, Rho5p, and Cdc42p (for review, see [6]). Of these, only Rho1p and Cdc42p are essential for cell survival. Cdc42p plays a major role in establishing cell polarity and morphology, whereas Rho1p is critical for cell integrity maintenance at every stage of growth [5]. Rho2p shares repair functions in cell integrity with Rho1p [8]. Rho3p and Rho4p regulate exocytosis, and both are necessary for cell separation [9,10,11], while Rho5p is the closest homologue of Rho1p and its elimination gives marginal phenotypes [12]. Fission yeast Rho GTPases are among the typical GTPases that cycle between an inactive GDP-bound conformation and an active GTP-bound conformation funnelling positive and negative stimuli into kinases, actin regulators, and adaptor proteins [13]. Because GDP is generally tightly bound and GTP is hydrolysed very slowly, small GTPases require guanine nucleotide exchange factors (GEFs), which facilitate GDP dissociation (needed for activation), and GTPase-activating proteins (GAPs), which increase the intrinsic GTP hydrolysis rate of the GTPases (needed for inactivation) [14,15,16]. The GDP-bound form of Rho proteins is recognised by guanine nucleotide dissociation inhibitors (GDIs), which sequester these proteins in the cytosol and prevent them from localising to membranes or being activated by GEFs (Figure 1A). The aim of this review is to provide an up-to-date overview of our understanding of Rho1p activators, Rgf1p, Rgf2p, and Rgf3p, focusing our attention on their role in the pathways involved in cell polarisation and cell integrity maintenance. In addition, we outline the unexpected roles of Rho1p-GEFs in signalling from compartments different from the plasma membrane (PM).

## 2. Structural–Functional Characteristics of Rho1p and Its GEFs Rgf1p, Rgf2p, and Rgf3p

*S. pombe* Rho1p (SpRho1p) is a functional homologue of human RhoAp (hRhoA) and budding yeast Rho1p (ScRho1p), sharing 65% identity with hRhoA and 73% identity with ScRho1p, as well as general structural features [17]. Rho1p bears the phosphate-binding loop (*P-loop*) (residues 14 to 21), two flexible regions, switch I (residues 29 to 39) and switch II (residues 62 to 79), which confer the guanine nucleotide-sensitive conformational changes, and the three-turn insertion helix (residues 125 to 139) involved in interactions with and the subsequent activation of downstream proteins. The switch from Rho1p-GDP to Rho1p-GTP is regulated by at least three GEFs, Rgf1p, Rgf2p, and Rgf3p [18,19,20,21,22], while GTP hydrolysis that returns the protein to the Rho1p-GDP conformation is stimulated by three GAPs, Rga1p, Rga5p, and Rga8p [23,24].

Rho1p-GEFs are multi-domain proteins that are much larger than Rho1p (Molecular weight (Mw) ~22 Kd), the subject of their activity. Rgf1p is the largest, with an Mw of ~150 Kd; Rgf3p has an Mw of ~144 Kd and Rgf2p has an Mw of ~130 Kd [25]. Rgf1-3p are structural orthologues of budding yeast Rho1p-GEFs, Rom1p, and Rom2p [26,27], and *Aspergillus fumigatus* Rom2p [28]. They belong to the family of DH-PH domain-containing Dbl (diffuse B-cell lymphoma)-related GEFs with more than 65 members in humans [29]. DH-PH GEFs share a catalytic Dbl homology (DH; also called RhoGEF) domain of 150–200 amino acids and an immediately adjacent regulatory pleckstrin homology (PH) domain of ~100 amino acids [30,31].

The DH domain can speed up the nucleotide exchange activity of Rho proteins by several orders of magnitude. DH binding induces conformational changes in the highly mobile switch regions of GTPase, promoting nucleotide ejection. The intermediate complex between the GEF and the nucleotide-free Rho protein does not accumulate in the cell. This is, in part, due to the high concentration of GTP and also because the binding affinity of nucleotide-free Rho protein is greater for GTP than for GEF proteins [14,15,32]. DH domains contain three conserved regions (CR1, CR2, and CR3) and form structures similar to elongated bundles of α-helices arranged in a “chaise longue” shape. Mutations within these conserved regions diminish nucleotide exchange activity [33]. This has been shown for Rgf3p in *S. pombe*, where a point mutation located on helix H8 (CR3) induces a hypomorphic and thermosensitive (ts) phenotype. Moreover, mutations in the same region of the Rgf1p and Rgf2p-DH domains also produce similar lack-of-function phenotypes [18,19,22].

PH domains in DH-PH Rho GEFs can promote or diminish nucleotide exchange via binding of the PH domain to phosphoinositides or through PH domain–protein interactions [34,35]. Occasionally, the tandem DH-PH domain can act as a “membrane-targeting device” by anchoring GEFs to the membrane (via phosphoinositides) and directing them towards their partner GTPases, which are on the membrane [36]. For instance, a ts mutation that falls between the PH and Citron and the NIK1-like kinase homology (CNH) domains of *rgf3*^+^ affected its localisation within the cell at the restrictive temperature [20]. Similarly, Rgf1p localisation to the division site was reduced in the *efr3*Δ mutant (a scaffold for the phosphatidylinositol-4 kinase Stt4p) [37] and in a temperature-sensitive mutant of the essential PI5-kinase *its3* (*its3-1*) [38]. These results suggest that Rgf1p membrane binding is PIP phosphatidylinositol (4,5)-bisphosphate (PI(4,5)P_2_)-sensitive. Furthermore, Rgf1p localisation is disrupted in the Rgf1pΔPH mutant lacking the PH domain [39].

Rho1p GEFs contain additional domains that enable their interaction with other proteins and surfaces. Rgf1p and Rgf2p have a DEP domain at the N-terminus, first discovered in flies (*Disheveled*), worms (*EGL-10*), and mammalians (*Pleckstrin*). DEPs are globular domains of about 100 residues with a high grade of sequence and structural similarity and different roles in signal transduction [40]. DEP domains are involved in plasma membrane anchoring [40], but they also participate in dimerisation [41], auto inhibition and phospholipid binding [42], and signal termination [43]. The DEP domain is present in Rgf1p and Rgf2p and is missing in Rgf3p. Rgf1p and Rgf2p localise both to the poles and the septum, while Rgf3p is cytoplasmic in the interphase and concentrates to the contractile acto-myosin ring (CAR) during cytokinesis [18,19,22]. However, removal of the DEP domain in Rgf1p (Rgf1pΔDEP) diminishes the signal at the poles and triggers the accumulation of the protein inside the nucleus [39]. This suggests a modest role for DEP domains in membrane tagging and uncovers their function to maintain the intramolecular interactions that drive changes in protein localisation (see the last section). The three Rho1p GEFs bear a C-terminal regulatory domain of 300 amino acids, termed the citron homology domain or CNH (from Citron-NIK-Homology). The CNH domain is present near the C-terminus of the Sticky/Citron kinase of flies, a RhoA-interacting kinase required for the organisation of the midbody in late cytokinesis [44] and in other kinases, such as the NIK (nck-interacting kinase) [45]. CNH domains are highly conserved in Rho1p-GEFs from fungi but are absent from RhoGEFs in higher eukaryotes [28]. Structurally, CNH domains belong to the super-family of β-propellers, with seven blades connected by small loops and arranged in a circular fashion [46]. However, to date, its function in the structure of kinases and fungal Rho GEFs is unknown. In *Aspergillus fumigatus,* the CNH domain of Rom2p interacts with the Rho1p Switch II motif and is required for cell wall synthesis [28]. The CNH domain of Rgf1p is required for its function in cell wall biosynthesis and as an upstream component of the cell integrity pathway [39]. In addition, the CNH domain of Rgf3p is required for viability, and substitution of the CNH domain of Rgf3p for the CNH domain of Rgf1p did not rescue lethality in the *rgf3*Δ/*rgf3*^+^ diploid (our unpublished data). These observations suggest that the CNH domain of Rgf1p and Rgf3p may facilitate different interactions providing some type of signal specificity. Our current view is that Rho1p GEFs contain N-terminal and C-terminal regulatory domains that maintain the GEF in an autoinhibited form by blocking the access of GTPases to the DH-PH domain. Accordingly, overexpression constructs lacking the N- or C-terminus of Rgf1p show different phenotypes, both being more severe than the overexpression of the full-length protein (our unpublished observations).

## 3. Rho1p Functions to Preserve Cell Integrity

The activity of Rho1p and its GEFs is tightly bound to the morphological transitions that run along the cell cycle progression. Newborn cells initiate monopolar growth, elongating from the “old end,” where the growth machinery is still concentrated, reminiscent of the previous septation. Then, at a point in G2, known as new end take off (NETO), new ends initiate growth, and both the microtubules and the actin cytoskeleton polarise towards both cell tips. When the cell reaches its maximal size, tip elongation ceases and mitosis occurs, followed by the assembly of a CAR that guides the division septum (Figure 1B) [47]. In each of these transitions, the addition of new membrane and cell wall material is prescriptive, and, at the same time, the cells must react to different stimuli without losing their viability [8].

Depletion of Rho1p activity causes cells to lyse during polar growth and cell division, with cytokinesis being the most critical step [18,48]. The cells shrink and die in a kind of “apoptosis” accompanied by the disappearance of polymerised actin, thus indicating that the main function of Rho1p is to preserve cell integrity. Intriguingly, Rho1p depletion is not suppressed by the addition of osmotic supplements that compensate for the leakage of a deficient cell wall, suggesting that Rho1p may perform other essential functions. These and other related questions remain unanswered. For instance, we ignore whether Rho1p activity is similarly regulated for septum assembly than for bipolar growth. Which are the cytoskeletal targets of Rho1p? How do different stimuli funnelled through Rho1p elicit different responses? In growing cells, Rho1p localises uniformly to the PM and internal membranes (our unpublished observations) and is slightly enriched at the growth tips (Figure 1C) [17,48]. At the onset of septum formation, the protein concentrates at future sites of cytokinesis and as the actomyosin ring shrinks, the Rho1p signal goes behind the ring, forming a disc that splits into two closely associated discs [17,21] (Figure 1C). Membrane localisation depends on a CAAX motif at the C-terminus that is geranyl-geranylated. In addition, Rho1p retention at cell tips is partially dependent on a phosphatidylserine gradient more negatively charged at the cell tip than on the cell sides [49]. However, given this nonspecific membrane localisation, GEFs and GAPs are essential to activate or inactivate Rho1p at the right time and the right place.

## 4. Connections between Rho1p and Rho1p GEFs and the Machinery That Determines Polar Growth

In *S. pombe*, the transition from monopolar to bipolar growth (NETO) is accompanied by cytoskeletal rearrangements. Before NETO, the polarity markers Tea1p and Tea4p are transported on growing microtubules plus ends to both cell tips [50,51,52,53,54], where they are released as discrete “dots” that anchor to the cortex through membrane proteins [55,56].

Upon NETO, the association of Tea1p and Tea4p promotes the binding of other polarity factors in large protein complexes at new ends. One of the factors recruited is For3p, which had before only been tethered to old ends [52]. Tea4p brings For3p into the proximity of formin activators, initiating actin cables assembly at the new end. In addition, the Tea1p–Tea4p complex recruits the kinase Pom1p, contributing to activation of Cdc42p as part of a positive feedback loop to maintain growth [47,57]. Accordingly, loss of Tea1p, Tea4p, For3p, or Pom1p impairs fission yeast polarisation [52,53,58]. In addition to actin cables, actin patches that guide endocytic vesicle internalisation also re-polarise to both cell tips upon NETO. It is known that the elimination of proteins involved in endocytosis also disturbs the establishment of polarity [59,60], suggesting that changes in protein composition at the cell tips are coupled with cytoskeletal rearrangements. Rho1p, Cdc42p, and Rho3p are key players in the regulation of the actin cytoskeleton (reviewed in [5,6,61]). However, the connections of Rho1p with the actin network have remained elusive, mainly because it is difficult to study the early functions of Rho1p in cells that frequently lyse. Rho1p is essential for viability, the spores deleted for Rho1p (*rho1::ura4*) are rounded, and their actin patches are delocalised, suggesting that Rho1p is required both for the localisation of polarised actin and spore outgrowth [17]. Depletion of Rho1p in growing cells occurs with a quick disappearance of polymerised actin and cell lysis, while its overexpression produces rounded cells and causes actin mislocalisation [17,48]. To date, Rho1p targets in the actin network are unknown. Preliminary work from our laboratory indicates that increased expression of Rho1p partially suppresses the thermosensitive phenotype of a formin mutant (*cdc12-112*), a tropomyosin mutant (*cdc8-117*), and a cofilin mutant (*adf1-1*) (our unpublished observations). Formins participate in the assembly of actin cables, and tropomyosin stabilises and increases the length of formin-nucleated actin filaments. In contrast, cofilin affects the severing of actin filaments, raising the rate of actin depolymerisation [62]. Thus, our data suggest that Rho1p regulates actin at different levels. This is not surprising given that actin structures (cables, patches, and rings) are required at all stages of growth. The limited knowledge we have about the role of Rho1p in polarised growth control comes from the study of one of its GEFs, Rgf1p [19]. Cells lacking Rgf1p are unable to activate/maintain bipolar growth and a fair number of them lyse at the growing end (Figure 2A). Accordingly, *rgf1*∆ cells are defective in actin organisation, with actin patches mostly present at only one end of the cell, the growing one. In addition, Rgf1p localisation is also dependent on the actin cytoskeleton, showing an interdependence in its location between Rgf1p and actin (unpublished results). Thus, Rgf1p participates in actin reorganisation required for bipolar growth activation [19]. This role is accomplished through Rho1p activation as cells bearing a mutation in the catalytic domain of Rgf1p with low exchange activity towards Rho1p [63] also show defects in bipolar growth (our unpublished data).

## 5. Role of Rho1p GEFs in Cell Wall Synthesis and Cell Integrity

Polarised growth is functionally linked to the actin cytoskeleton, which provides the infrastructure and materials for PM and cell wall synthesis. For instance, in experiments on the regeneration of protoplasts (spherical cells where the wall has been removed), the deposition of new wall material depends on the actin cytoskeleton [64]. The fission yeast cell wall is a polysaccharide matrix that resists internal turgor pressure and protects against different types of injury. Under the electron microscope, the cell wall is viewed as a central electron-transparent layer filled with α- and β-glucans sandwiched between two thin electron-dense layers formed by galactomannoproteins [65]. β-glucan provides strength to the cell wall and is the first polysaccharide deposited on the newly forming wall during protoplast regeneration and the spore wall [66,67]. Despite the existence of different types of β-glucans: linear β-(1,3)-glucan, branched β-(1,3)-glucan, and branched β-(1,6)-glucan [68,69], only β(1,3)-glucan synthase activity (βGS) has been described to date. This enzyme is associated with the inner surface of the PM and uses UDP-glucose as a substrate, forming linear glucose chains. Based on the work of Cabib and co-workers, who developed an in vitro system for β-glucan synthesis, the βGS activity was shown to function as a complex of at least two components: a catalytic fraction and a regulatory one [70]. Fission yeast Rho1p acts as the regulatory subunit β–GS complex in combination with four interchangeable catalytic subunits, Bgs1p, Bgs2p, Bgs3p, and Bgs4p [66,71,72,73,74,75]. However, to date, only mutants in Bgs4p display reduced levels of cell wall β-glucan and β–glucan synthase (GS) activity [71]. Rho1p directly stimulates GS and glucan synthesis in its GTP-bound prenylated form, allowing the cell to switch β(1,3)-glucan synthesis on and off by interconverting the GDP and GTP forms of Rho1p [76]. In this task, Rho1p activity is stimulated by three GEFs, Rgf1p, Rgf2p, and Rgf3p, which act at different times and places throughout the yeast life cycle [18,19,20,21,22,77]. These three play important roles in regulating cell wall synthesis.

Rgf3p is the only essential regulator of Rho1p identified thus far. *rgf3*^+^ was cloned by complementation of the *ehs2-1* mutant (echinocandin-hypersensitive) that is very sensitive to drugs that interfere with cell wall biosynthesis [22]. Rgf3p depletion causes cell lysis during cytokinesis, a phenotype also seen in cells lacking Rho1p or Pck1/2p activity (Figure 2C) [78]. Rgf3p exclusively localises to the septum area (Figure 1C) and activates GS, increasing the amount of cell wall β-(1,3)-glucan [22]. Thus, Rgf3p may stimulate Rho1p-mediated activation of glucan synthase activity, which is necessary for septum functioning and cell separation (see the cytokinesis section).

Rgf1p was identified as a Rho1p GEF required for organising actin deposition coupled to cell wall biosynthesis [19]. Rgf1p acts as a β-GS activator in a Rho1p-dependent manner. First, depletion of Rgf1p causes hypersensitivity to Caspofungin, a drug that inhibits β(1,3)-glucan synthesis [79]. *rgf1*∆ cells lyse at one of the poles mainly during the activation of the bipolar growth, with a phenotype similar to the cells expressing the Rho1T20N-dominant negative mutant (Figure 2A) [17,19]. In addition, *rgf1*^+^ overexpression raises the amount of GTP-bound Rho1p and increases β-GS activity by severalfold, generating a massive accumulation of cell wall material [19]. Caspofungin sensitivity in *rgf1*∆ cells is suppressed by mild overexpression of the β-GS catalytic subunit Bgs4p, suggesting that Rgf1p specifically activates the Rho1p-Bgs4p GS complex. Interestingly, *rgf1*Δ cells are severely defective in cell wall (CW) mechanical homeostasis [80]. It is likely that rapid growth concomitant with rapid CW thinning eventually causes tip lysis and cell death in *rgf1*∆ cells.

Despite not being essential, Rgf1p is the main activator of Rho1p during vegetative growth [81]. Rgf2p compensates for the lack of function of Rgf1p during polar extension [18]. Rgf2p also interacts with Rho1p [21], and although *rgf2*∆ cells are phenotypically indistinguishable from wild-type cells, the double mutant *rgf1*∆*rgf2*∆ is not viable. Results from our laboratory demonstrate that *rgf1-45*Δ cells (with a deletion of 45 aa at the C-end) can grow at 32 °C, while the *rgf1-45*Δ *rgf2*Δ mutant undergoes explosive lysis in early G2 at the same temperature (our unpublished observations) [7]. However, the best-known function of Rgf2p takes place during sexual differentiation. The *rgf2*^+^ mRNA is highly induced in sporulation (after meiosis II) [18,82]. Accordingly, Rgf2p-GFP is hardly seen in vegetative cells, but it appears clearly around the spore’s membrane (Figure 1C). Rgf2-null mutant homozygous crosses produce spores with a dark and immature appearance (Figure 2B). This phenotype was almost identical to that seen in spores deleted for *bgs2*^+^, the sporulation-specific β-glucan synthase subunit [66]. In fact, sporulating *rgf2*Δ diploids show very little β-GS activity. Thus, Rgf2p plays an essential role in activating the module Rho1p-Bgs2p during spore wall formation [18]. In summary, Rho1p and Rho1p GEFs regulate the synthesis of β-glucan when and where CW remodelling is required: growth by the poles, division, and spore formation.

In addition to cell cycle control, cell wall assembly and cellular integrity are constantly challenged by variations in intra- and extracellular conditions. Cells rely on signalling pathways to respond and adapt to these changes [8,83]. Typically, specific sensors at CW and PM detect extracellular stimuli and transmit a signal to a MAPK cascade that, in turn, activates the effectors.

Rho1pGEFs are connected to cell wall sensors Wsc1p and Mtl2p [84,85] and to Rho1p, which, in turn, regulates cell integrity directly through its interaction with the β(1,3)-glucan synthase and indirectly by the effector kinases Pck1p and Pck2p (the orthologues of *Saccharomyces cerevisiae* Pkc1p and human PKC). Pck1p and Pck2p share an essential function in cell viability and regulate the synthesis of α- and β-glucans, which are the main components of the cell wall [8]. In addition, Rho1p, Pck1p, and Pck2p act upstream of the mitogen-activated protein kinase (MAPK) module of the cell integrity pathway (CIP) that includes Mkh1p (MAPKKK), Skh1p/Pek1p (MAPKK), and Pmk1p/Spm1p (MAPK) [63,86,87,88,89]. The CIP becomes activated by CW damage and saline and osmotic stress [90] and regulates cell wall remodelling, cell separation, and ion homeostasis in injured cells (revised in [8,91]). Rho1p controls Pmk1p basal activity during vegetative growth mainly through Pck2p [89], while Rgf1p acts as a bona fide activator of the CIP in response to cell wall damage and osmotic shock [63]. The *rgf1*-null mutant shows phenotypes also shown in mutants deleted for MAPK components: the *vic* phenotype (viable in the presence of an immunosuppressant and chlorine ion) and sensitivity to high salt concentrations. In addition, deletion of Rgf1p strongly diminishes Pmk1p phosphorylation (activation) upon osmotic shock and cell wall damage. Interestingly, Rho1p and Pck2p are both involved in Pmk1p activation mediated by Rgf1p, but not the other two Rho1p-specific GEFs, Rgf2p and Rgf3p [63]. CIP activation requires accurate regulation, and while proper activation of the CIP pathway is positive for maintaining cellular integrity, its constitutive activation observed in a *rho1-596* thermosensitive strain is detrimental to the cell [92].

## 6. Role of Rho1p GEFs in Cytokinesis

Cells devoid of Rho1p activity die in pairs of shrinking cells, mainly during cytokinesis, suggesting that Rho1p is critical at that point of the cell cycle. Interestingly, the three GEFs (Rgf1p, Rgf2p, and Rgf3p) localise to the medial region with slightly different timings and positions. To pinpoint the time of appearance of each GEF, we review the main steps of the cytokinesis processes (Figure 3) [93,94,95]. Early cytokinesis starts with the selection of the division site, which is determined by a band of precursor nodes localised to the cortex near the nucleus [96,97]. Upon mitosis entry, contractile-ring assembly begins when Mid1, the anillin-like protein, recruits other proteins to assemble the cytokinesis nodes that will, later on, condense and collapse into a ring. These proteins include the Myosin II heavy and light chains, the IQGAP Rng2p, the F-BAR protein Cdc15p, and the formin Cdc12p, among others (Assembly, Figure 3) [98,99]. During maturation (Maturation; Figure 3), the ring is maintained (without contraction) until the end of the anaphase. Once the sister chromatids are segregated on both sides of the contractile ring, progressive CAR constriction guides ingression of a membrane and a walled septum that prepares cytokinesis for cell separation (Constriction/Septation; Figure 3) [100,101]. Of the three GEFs, Rgf3p localises exclusively to the contractile ring (Figure 1C) [20,21,22]. Rgf3p-GFP first appears in the early anaphase when SPBs are ~3 μm apart and contracts with the ring until the signal becomes a dot and disappears (Figure 1C). Super-resolution microscopy studies place Rgf3p in an intermediate layer of the ring that includes paxilin Pxl1p, the cytokinetic factor Fic1p, Rho GAP Rga7p, the kinase Pck1p, the phosphatase Clp1p, the DIRK kinase Pom1p, and the nebulin family protein Cyk3. This layer is surrounded by a membrane proximal layer composed of membrane-bound scaffolds such as Mid1, Cdc15, and Imp2 F-BAR proteins and by F-actin and the motor domains of myosins type II and V on the other side, farthest from the membrane [102]. The localisation of Rgf3p to pre-constriction rings depends on Cdc15p and Imp2p, which recruit Pxl1p, Fic1p, and Cyk3p through their SH3 domain [103,104,105], and is independent of actin filaments [21,106]. In addition, the arrestin family protein Art1p reduces the localisation of Rgf3p to the ring and its protein level, suggesting a role for Art1p connecting Rgf3p to receptors of the endocytic machinery [106]. The localisation of Rgf1p is quite different. The medial Rgf1p signal appeared during the anaphase before the emergence of the primary septum (PS). The Rgf1p signal concentrates in a ring that moved centripetally with the growing septum (Figure 1C). In co-localisation experiments, Rgf1p follows the Rlc1p (an integral component of the ring) signal, localising at the advancing septum edge and leaving behind a fluorescent trace as the cell wall of the septum grows towards the cell interior [107]. This means that the Rgf1p ring is located outside the contractile ring. Apparently, the localisation of Rgf2p is similar to that of Rgf1p [18,21]; however, revisiting Rgf2p temporal and spatial localisation with better microscopes may reveal some differences in the future.

## 7. Rgf3p Plays an Essential Function in Septum Synthesis

As pointed out previously, the latest stages of cytokinesis require the assembly and constriction of a walled septum. Under electron microscopy, the septum is perceived as a laminar structure in with the primary septum (PS) layer is sandwiched between the two secondary septum layers (SS) [68]. In this setup, the PS is degraded and disappears during cell separation while the SS forms the new cell wall of the just divided cells. The PS is made of linear-β-1,3-glucan synthesised by glucan synthase Bgs1p/Cps1p and contains branched-β-1,3-glucan. The secondary septum (SS) consists of 1,6 branched β-1,3-glucans synthesised by Bgs4p and α-1,3-glucans synthesised by Ags1p and β-1,6-glucans [100,101]. β–Glucan synthesis is carried out by the β–GS complex with Rho1p as the regulatory subunit and four catalytic subunits, Bgs1p, Bgs2p, Bgs3p, and Bgs4p [1,100]. Except for Bgs2p, the other Bgss appear at the septum membrane. Bgs1p localises as a ring tightly associated with CAR and with the septum membrane during ingression (Figure 1). Bgs3p and Bgs4p follow CAR contraction but remain localised as a disc along the invaginated PM; Bgs4p is essential for SS formation and proper PS completion, while the Bgs3p function has not yet been assigned [1,100]. The three Bgss must be activated/inactivated in place and in time by Rho1p signalling. However, it is unclear whether a single GEF regulates β–GS activity during PS synthesis and whether a different one activates β–GS for SS synthesis.

Biochemical evidence for the selective action of Rgf3p on Rho1p has been shown [22]. Rgf3p expression peaks during septation in an Ace2p-dependent manner and the protein is phosphorylated [20,108]. To our knowledge, Rgf3p is the main candidate for the role of a positive regulator of Rho1p during septum synthesis and likely for PS synthesis. First, the localisation of Rgf3p mimics the localisation of Bgs1p (in charge of PS synthesis). Rgf3p could act as a physical link between components of the CAR and membrane-bound Bgs-mediated septum growth [102]. As mentioned previously, CAR-localised proteins, such as Cdc15p, Imp2p, and Art1p, recruit Rgf3p [103,106]. Cdc15p also participates in the transport of Bgs1p to the septum membrane [109,110]; therefore, it is possible that the concerted action of these proteins regulates the traffic of Bgs1p to the PM while activating the regulatory subunit of β-GS. Second, the connection between Rgf3p and cell wall integrity maintenance is supported by several observations. Cells depleted of *rgf3* lyse as pairs during cell separation, mimicking the phenotype of *rho*-depleted mutants. None of the other GEFs (Rgf1p and Rgf2p) showed this type of lysis. Mild overexpression of Bgs1p, Bgs2p, and Bgs3p but not of Ags1p (the α-GS) in multi-copy plasmids suppresses the echinocandin hypersensitivity in *ehs2-1* cells (a mutant allele of *rgf3*) [22]. Third, a different allele of *rgf3*^+^, *lad1-1,* undergoes cell lysis specifically during cell division, very similar to that seen in the *ehs2-1* allele. Electron microscopy analysis of *lad1-1* cells indicates that lysis occurs only as the primary septum begins to degrade [20]. Accordingly, the ts- and lytic phenotype displayed by the *ehs2-1* mutant is suppressed by elimination of β-glucanase, which specifically degrades the PS β-glucan (Eng1p) [111] but not the α-glucanase (Agn1p) in charge of the degradation of the wall material that surrounds the septum [112,113] (our unpublished data).

The role of Rgf3p at the early stages of cell division have been neglected, in part, because the strong lytic phenotype of the *rgf3* mutants and, in part, because the other two GEFs, Rgf1p and Rgf2p, also localise at the division site. We have shown that mild expression of *rgf1*^+^ (in a multi-copy plasmid and under the control of its own promoter) cannot overcome the lysis phenotype of *ehs2-1* cells at 37 °C [22]. In addition, the elimination of *rgf1*^+^ in the *ehs2-1* mutant background produces viable cells at 28 °C but not at 37 °C, a temperature that allows both mutants to grow on plates [19]. Both results suggest that Rgf1p and Rgf3p are not functionally exchangeable. However, it still needs to be determined whether cell death in the *rgf1*Δ *ehs2-1* occurs during cell separation, or if it is a consequence of the sum of tip growth defects plus septation defects. Regarding Rgf2p, cells of the double *rgf2*Δ *ehs2-1* mutant are viable at all temperatures and are phenotypically similar to *ehs2-1* cells.

## 8. Rgf1p Is Involved in a Cytokinesis Checkpoint Together with the Septation Initiation Network (SIN) and the Cell Integrity Pathway (CIP)

Another point that remains uncertain is the relationship between Rho1pGEFs and the septation initiation network (SIN), the signalling pathway that controls the timing of cytokinesis, guaranteeing that chromosomes are segregated on both sides of the CAR. The SIN is required for ring maintenance; in consequence, insufficient SIN signalling generates multinucleated cells without rings or septa [114,115]. This signalling cascade monitors the position of the spindle pole bodies (SPBs) (the yeasts equivalent to centrosomes) throughout the cell cycle to coordinate nuclear/spindle positioning with the exit from mitosis and the onset of cytokinesis [116]. The SIN senses SPB positioning through a Ras-like GTPase, Spg1p, whose activation is controlled by a bipartite GAP, a scaffold complex that anchors the pathway to the cytoplasmic side of the SPBs and a linear cascade of three kinases (Cdc7p, Sid1p, and Sid2p) [115]. Despite our knowledge of the SIN pathway, it is still unknown how the SIN transmits SPB positioning to the CAR to activate septum assembly. Sid2p kinase (a nuclear Dbf2-related (NDR) kinase) plays a critical role in this task. Out of the thirteen essential SIN components, only Sid2p and its counterpart Mob1p are associated with the CAR in the mid-late anaphase [117]. In addition, all SIN substrates identified thus far are Sid2p substrates. These include the phosphatase Clp1p, the formin Cdc12p, the kinesin Klp2p, the SAD kinase Cdr2p, and the anillin-like Mid1p [118,119,120,121,122]. However, no SIN targets have been detected among the cell wall biosynthetic enzymes, partly because proper localisation of these enzymes requires an intact actomyosin ring [71,72] and the SIN mutants do not form stable rings [123]. Interestingly, upregulation of Rho1p and Rgf3p suppresses the ts phenotype of *sid2* mutants at a semi-restrictive temperature [124]. In addition, it has been proposed that Rho1p is involved in feedback activation of Spg1p during actomyosin ring constriction, ensuring SIN activity while cytokinesis is in progress [125].

Rgf1p does not seem to activate Rho1p for SIN signalling. *rgf1*Δ early anaphase cells contain regular acto-myosin rings, and cytokinesis timing is almost identical in cells with or without Rgf1p. However, there is a connection between Rgf1p and the SIN pathway. Work from our laboratory shows that cell wall damage inflicted during cytokinesis triggers a checkpoint-like response, promoting a delay that occurs right before ring constriction (more precisely during ring maturation) [107]. This delay depends on Rgf1p/Rho1p and Pck2p and is abolished in the absence of MAP kinase of the cell integrity pathway (CIP). Because inactivation of this pathway in stressed cells causes defects in septation [126,127], cell integrity pathway (CIP) signalling may delay ring constriction in response to cell wall perturbations to ensure that cytokinesis reaches completion only after the cell has adjusted to the new conditions. The delay induced by the wall-damaged cytokinesis checkpoint correlates with a prolonged SIN signal and depends on the SIN to be achieved. Given that Sid2p is required for actomyosin ring maintenance when the cytokinesis checkpoint is active, it is likely that prolonged SIN activity serves to maintain the ring in a competent state to achieve constriction safely under cell wall stress [107]. In addition to defects in cytokinesis, deletion of *rgf1*^+^ results in an abnormally high percentage of cells that grow only at one end (monopolar cells). Interestingly, mutants of Fic1p, Imp2p, the septin Spn1p, and a number of proteins affecting the last steps of cytokinesis also exhibit new end growth polarity errors (NETO defects), suggesting that successful completion of cell division that cleans out cytokinesis factors contributes to a new end-growth competency [128,129].

## 9. Role of Rho1p GEFs in Genomic Instability

In addition to their classical role as membrane-bound signal-transducing molecules, Rho GEFs, Rho GTPases, and downstream components localise to the nuclear membrane and inside the nucleus, suggesting that Rho-related signalling processes may also take place in this cellular compartment [130]. Moreover, actin is constantly shuttling between the nucleus and cytoplasm [131]. It was thought for a long time that nuclear actin was monomeric actin, principally involved in transcriptional regulation through transcription factors, chromatin-remodelling complexes [132], and RNA polymerases [131]. More recently, the development of fluorescently tagged actin-specific probes for live imaging of actin filaments has implicated transitory nuclear actin polymerisation in chromatin organisation at the mitotic exit, in response to serum stimulation and in DNA repair mechanisms [133,134,135]. However, little is known about the regulators or the mechanisms by which actin organises nuclear content in different situations. As pointed out previously, Rgf1p concentrates in the cell nucleus during a replication blockage triggered by hydroxyurea (HU) and its accumulation is required for chronic tolerance to the drug [39]. HU reduces intracellular pools of deoxyribonucleoside triphosphate by inhibiting ribonucleotide reductase and blocks DNA synthesis, activating the DNA replication checkpoint pathway. Fission yeast has two genetically distinct checkpoint-signalling pathways that respond to DNA damage: the replication checkpoint, mediated by Cds1p kinase and activated in response to stalled replication forks and DNA damage occurring in the S phase, and the G2/M checkpoint, mediated by Chk1p kinase, which responds to strand breaks and other types of damage that may occur during the G2 phase [136]. Rgf1p nuclear accumulation during replication arrest depends on the 14-3-3 chaperone Rad24p and the DNA replication checkpoint kinase Cds1p. Both proteins control the nuclear accumulation of Rgf1p by inhibition of its nuclear export. When cells are subject to replication stress, Rgf1p changes its conformation, probably by Cds1p phosphorylation, which allows its interaction with Rad24p. This remodelling would hide the NES sequence, reducing its association with Crm1p and, thus, blocking its exit from the nucleus [39]. The role of Rgf1p under replicative stress is unclear. We have shown that an Rgf1p mutant, Rgf1p-9A, in which the serines of the nine potential Cds1p phosphosites (RXXS) have been substituted with alanines, does not interact with endogenous Rad24p. The Rgf1p-9A mutant fails to accumulate in the nucleus in response to replication stress and displays a severe defect in survival in the presence of HU. Interestingly, mutant cells do not show the phenotypes characteristic of the *rgf1*Δ cells, such as monopolar growth, sensitivity to caspofungin, and the *vic* phenotype, suggesting that mutation of the Cds1p phosphosites solely affects HU survival [19,63]. Therefore, the interaction of Rgf1p-Cds1p-Rad24p is required especially for tolerance to replication stress, suggesting that Rgf1p could be part of the mechanism by which Cds1p promotes survival [137]. However, later on, we found that Rgf1p is also involved in tolerance to genotoxic agents such as camptothecin (CPT, a topoisomerase inhibitor) and phleomycin (Phl, a derivative of bleomycin); both agents induce DNA double-strand breaks (DSBs). In yeasts, which repair DSBs primarily by homology-directed repair (HDR), the induction of a single chromosomal break triggers increased local mobility. HDR initiates when the broken chromosomal ends are resected by nucleases and helicases, generating 3′ single-stranded DNA (ssDNA) overhangs onto which the Rad51p recombinase assembles as a nucleoprotein filament. This structure is able to invade a homologous duplex DNA strand and serve as a primer for copy synthesis [138,139]. Moreover, multiple DSBs form clusters (also named foci), and clustering may facilitate homology searching and increase repair efficiency. Our laboratory has shown that Rgf1p is involved in the repair of DNA DSBs induced by Phl treatment [7]. The Rgf1p mutant cells are defective in reentry into the cell cycle following the induction of severe DNA damage. This phenotype correlates with the inability of *rgf1*Δ cells to repair fragmented chromosomes together with its inability to remove the Phl-induced Rad52p-YFP foci. These phenotypes are phenocopied by genetic inhibition of Rho1p [7]. The role of actin polymerisation in DSB repair has not been characterised; however, recent findings in *Xenopus* eggs support a model for HDR DNA DSB repair in which the Arp2/3 complex and its activator, WASP, bind at the break and work with actin to promote DSB clustering and HDR [140]. Finally, to delineate the role of Rgf1p/Rho1p over the Rad52p factories that propitiate DSB repairs, it will be of interest to know whether these proteins are involved in clustering mobility.

## 10. Concluding Remarks

This review highlights the participation of Rho1p in distinct cellular processes depending on the stimulus and cell cycle stage (Figure 4). This is due to the tight regulation carried out by GEFs and GAPs that determine its activation and inactivation in place and time. In this way, Rho1p regulates cell wall synthesis whenever this structure needs to be remodelled: at the growing pole, during septation, and in spore maturation (Figure 4E,F). Moreover, it guarantees cellular homeostasis when CW and membrane reorganisation at the growth zones is a challenge to the cell and (by activating the CIP) when environmental conditions put it in danger (Figure 4B,C). In addition, Rgf1p-dependent Rho1p activity participates in the control of polarised growth, probably achieved by maintaining the local accumulation of polarity factors and key actin proteins at cell growth sites (Figure 4A). Nevertheless, Rho1p not only carries out its activity in cell boundaries but also may perform functions onto the nuclear periphery. Through its regulation by Rgf1p, Rho1p participates in the repair of DSBs and, therefore, in the maintenance of genomic stability (Figure 4D). GAPs are also important in maintaining a balanced Rho1p activity. According to its involvement in negative regulation of Rho1p, the elimination of Rga1p produces cells with abundant actin patches and a thick cell wall, while the elimination of Rga5p causes a mild increase in cell wall biosynthesis and a mild defect in septation.

In addition, signalling by Rho GTPases is controlled at the level of gene expression and by post-transcriptional modifications. In this sense, phosphorylation could be one of the most likely modifications for both Rho1p and its regulators. Phosphoproteomic studies have detected possible phosphorylation sites at the three GEFs of Rho1p [141]. However, little is known about the kinases responsible for this phosphorylation. Future research will be required to elucidate the effect of these modifications on the regulation of Rho1p and its GEFs and the comprehension of their functional relevance.

## Figures and Tables

**Figure 1 ijms-23-13888-f001:**
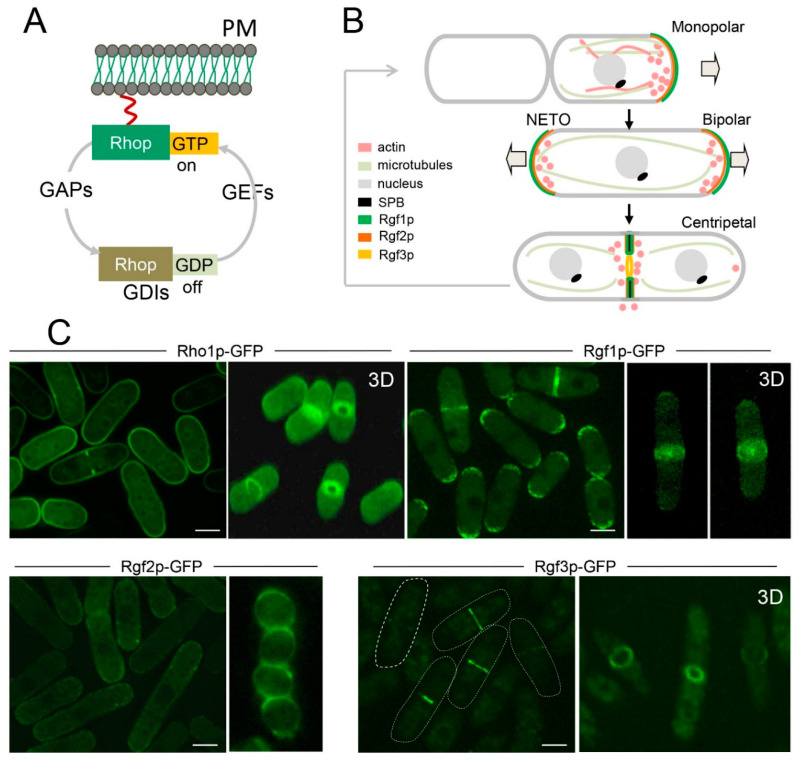
(**A**). Rho GTPase regulation. Rho GTPases cycle between GDP-bound inactive and GTP-bound active states. They are activated by GDP/GTP exchange stimulated by GEFs and inactivated by GTP hydrolysis stimulated by GAPs. Some members also showed a cytosol/membrane alternation regulated by GDIs. (**B**) The cell cycle of fission yeast. Newborn cells initiate monopolar growth, elongating from the “old end.” During the interphase, cells grow from both cell tips (grey arrows) to their maximal length (~14 µm). During mitosis, the mitotic spindle segregates the chromosomes and the cell assembles an actomyosin contractile ring (yellow) between the poles of the spindle. The contractile ring constricts in cytokinesis guiding the division septum (black). (**C**) Maximum projection images of vegetative cells expressing Rho1p-GFP, Rgf1p-GFP, and Rgf3p-GFP endogenously. Rgf2p-GFP was visualised in cells transformed with a plasmid (pAL-*rgf2*^+^-GFP). Cells were grown at 28 °C to the early log phase in YES medium. The panels on the right show the 3-D reconstruction of Rho1p-GFP, Rgf1p-GFP, and Rgf3p-GFP localisation to the medial rig during septum formation. Spores expressing Rgf2p-GFP endogenously show fluorescence over the spore wall (middle panel). Scale bars represent 4 µm.

**Figure 2 ijms-23-13888-f002:**
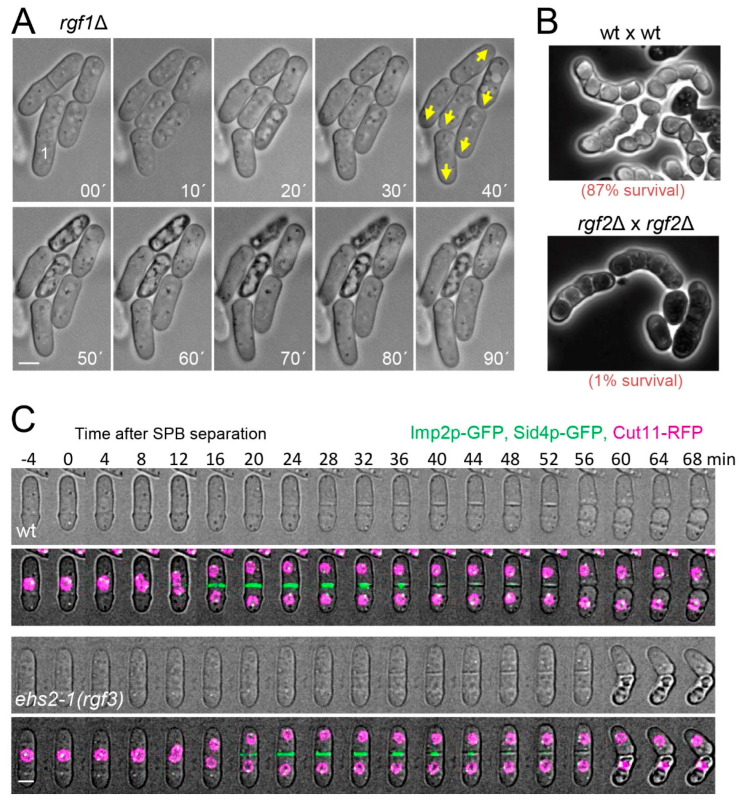
Cell integrity defects in Rgf1p, Rgf2p, and Rgf3 mutants. (**A**) Time-lapse DIC images of *rgf1*Δ cells showing different growth patterns. Yellow arrows indicate the direction of growth. The siblings of cell 1 show an abnormal growth pattern; both grew in the same direction: one daughter from its old end and the other from its new end. The one that grew in the wrong direction immediately underwent lysis. (**B**) Sporulation phenotype of wild type crosses (*rgf2*^+^ h^+^ × *rgf2*^+^ h^−^) and *rgf2*Δ crosses (*rgf2*Δ h^+^ × *rgf2*Δ h^−^). Cells of the genotypes indicated were incubated for 48 h on MEA at 28 °C. Phase contrast micrographs are shown. (**C**) Time lapse series of wild type and *ehs2-1* (*rgf3*) cells expressing Imp2p-GFP (CAR marker), Sid4p-mCherry (SPB marker), and Cut11p-RFP (nuclear membrane marker) grown at 28 °C. The images shown are the maximum-intensity projections of z stacks. The fluorescence images are shown in the lower panels. Scale bars represent 3 µm.

**Figure 3 ijms-23-13888-f003:**
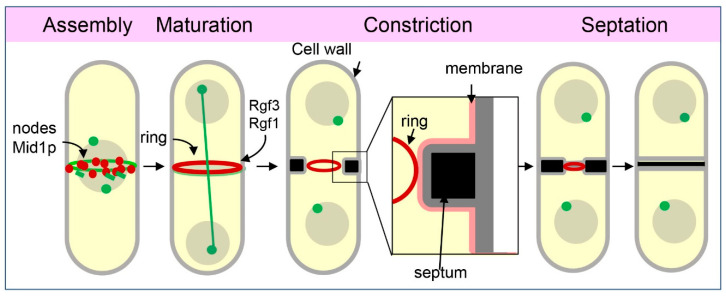
Schematic representation of fission yeast cytokinesis. In the assembly step, the cytokinetic nodes coalesce into a ring with homogeneous fluorescence. Once the ring is assembled, a maturation step with interchanges between cytoplasmic and ring proteins takes place. Rgf1p and Rgf3p first appear to the ring in this step, which lasts approximately 10 min. Cytokinesis continues with ring constriction, which reduces the circumference of the ring until its complete closure, and it is coupled with septation. The ring forms under the surface of the PM and is linked to the PM such that, when it constricts, it guides the ingression of a walled septum that partitions the cell in two.

**Figure 4 ijms-23-13888-f004:**
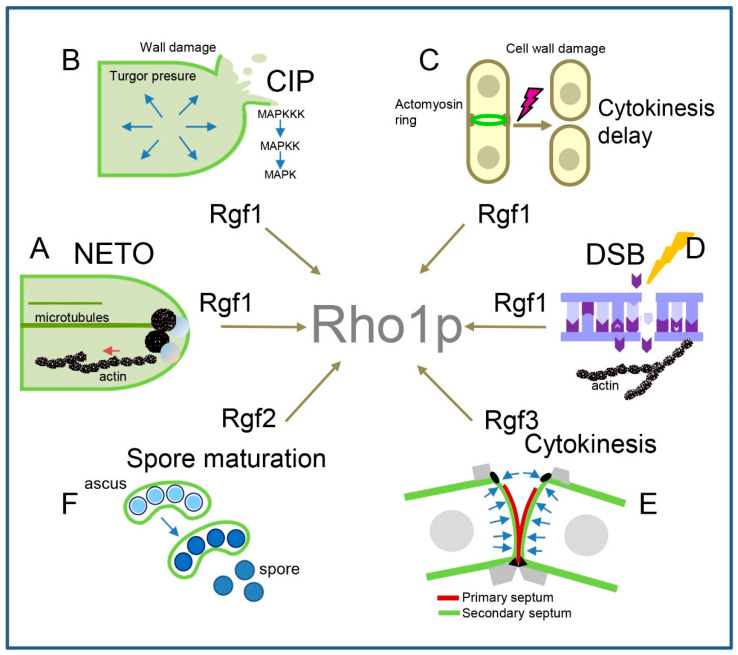
The various functions of Rho1p in the life cycle of fission yeast. (**A**) Rgf1p-dependent Rho1p activity participates in the activation of bipolar actin and bipolar growth during NETO. (**B**) Rgf1p induction of Rho1p activity is required to maintain cell homeostasis in response to cell wall damage and osmotic shock signalling through CIP. (**C**) Rgf1p and Rho1p participate in a checkpoint pathway that controls actomyosin ring constriction in response to cell wall damage. (**D**) Rgf1p and Rho1p activity positively controls the repair function that confers resistance against the anti-cancer drug Phleomycin, which induces double-strand breaks in the DNA. (**E**) Rgf3p induction of Rho1p activity during cytokinesis promotes cell wall remodelling, which is essential to achieve cell separation. (**F**) Rgf2p-specific Rho1p activity plays an essential function in the synthesis of the β–glucan layer necessary to achieve spore maturation.

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
