# Peer review of "Fission Yeast Rho1p-GEFs: From Polarity and Cell Wall Synthesis to Genome Stability"

_ijms, 2022, doi:10.3390/ijms232213888_

Round 1
Reviewer 1 Report
Dear Editor,
Thank you for the opportunity to review this manuscript, dealing with interesting findings entitled “Fission Yeast Rho1p-GEFs: From Polarity and Cell Wall Synthesis to Genome Stability”. This review focuses on fission yeast Rho1p GEFs, Rgf1p, 15 Rgf2p, and Rgf3p that belong to the family of DH-PH domain containing Dbl-related GEFs. In addition, they outline the non-canonical roles of Rho1p-GEFs in genomic stability. All the explanations are interesting, and the article includes a balanced and critical view of the outcomes. All figures' quality is good with significant explanations in the text material as well as figure legends. In addition, the authors have cited appropriate and adequate references to related and previous work. Therefore, the review article entitled “Fission Yeast Rho1p-GEFs: From Polarity and Cell Wall Synthesis to Genome Stability” could be published in the International Journal of Molecular Sciences in its current stage.
Author Response
Thank you for your comments
Reviewer 2 Report
This is a nice review focusing on Rho1 functions and regulation by GEPs (Rgf1, Rgf2. Rgf3) in fission yeast. The manuscript covered various aspects of Rho1 functions and related cellular events including polar growth, cytokinesis, septum formation, genome stability and spore formation. It is nicely summarized and well written.
Some minor comments.
It is not much stated about the role of Rho1 GAPs (Rga1, Rga5, Rga8). The summary slide Figure 4 only showed the role of GEF in different events. This might give an impression Rho1 is regulated exclusively by GEP not by GAP. It is not clear how different GAPs are involved in these events. I fell it is better to put at least some comments on GAPs in this summary slides.
Line199 and other places. Bold style letters are used, they may be not necessary.
Author Response
We have addressed point by point the questions posed by the editorial and the referees. I would like to thank the editor and the referees for their comments.
- Regarding the referee 2 comments.
- Line199 and other places, bold style letters are used, they may be not necessary.
I think that this may be an editorial problem because in the word version that was sent to the journal, the Line 199 does not contain bold type letters.
- It is not much stated about the role of Rho1 GAPs (Rga1, Rga5, Rga8). The summary slide Figure 4 only showed the role of GEF in different events. This might give an impression Rho1 is regulated exclusively by GEP not by GAP. It is not clear how different GAPs are involved in these events. I fell it is better to put at least some comments on GAPs in this summary slides.
Regarding the function of Rho1p GAPs, we agree with the referee comments and we have added a paragraph at the Concluding remarks section. “GAPs are also important in maintaining a balanced Rho1p activity. According to its involvement in negative regulation of Rho1p, the elimination of Rga1p produces cells with abundant actin patches and a thick cell wall while the elimination of Rga5p causes a mild increase in cell wall biosynthesis and a mild defect in septation”. However, we have not changed the summary slide (Figure 4), because our knowledge about Rho1GAPs function is a broad-spectrum function (increase in cell wall biosynthesis, actin disorganization…etc), and it does not fit on the more specific processes described in Figure 4 (the DNA damage response, sporulation and NETO..etc). In addition, this review concerns GEFs activity and we believe that the summary slide should focus the information on GEFs not adding misunderstanding to the take home message.
